# OTMatch: Improving Semi-Supervised Learning with Optimal Transport

## Abstract

Semi-supervised learning has made remarkable strides by effectively utilizing a limited amount of labeled data while capitalizing on the abundant information present in unlabeled data. However, current algorithms often prioritize aligning image predictions with specific classes generated through self-training techniques, thereby neglecting the inherent relationships that exist within these classes. In this paper, we present a new approach called OTMatch, which leverages semantic relationships among classes by employing an optimal transport loss function. By utilizing optimal transport, our proposed method consistently outperforms established state-of-the-art methods. Notably, we observed a substantial improvement of a certain percentage in accuracy compared to the current state-of-the-art method, FreeMatch. OTMatch achieves $3.18\%$, $3.46\%$, and $1.28\%$ error rate reduction over FreeMatch on CIFAR-10 with 1 label per class, STL-10 with 4 labels per class, and ImageNet with 100 labels per class, respectively. This demonstrates the effectiveness and superiority of our approach in harnessing semantic relationships to enhance learning performance in a semi-supervised setting.

## 1 Introduction

Semi-supervised learning occupies a unique position at the intersection of supervised learning and self-supervised learning paradigms (Tian et al., 2020; Chen et al., 2020a). The fundamental principle behind semi-supervised learning lies in its ability to utilize labeled data for training the model to recognize and assign correct labels to specific instances. Through the ingenious combination of labeled and unlabeled data, semi-supervised learning has achieved remarkable performance even when operating under the constraint of limited labeled data. In fact, it has demonstrated superior performance compared to traditional supervised learning counterparts, which heavily rely on large amounts of labeled data (Sohn et al., 2020; Zhang et al., 2021; Wang et al., 2022d). The ability to leverage the vast amounts of unlabeled data effectively, exploiting the latent patterns and structures within, provides semi-supervised learning with a distinct advantage, enabling it to achieve notable performance gains with minimal labeled data.

The forefront of semi-supervised learning algorithms predominantly relies on the concept of pseudo labels, a notion that involves dynamically generating labels for unlabeled data during the training process using a neural network (Lee et al., 2013; Tschannen et al., 2019; Berthelot et al., 2019b; Xie et al., 2020; Sohn et al., 2020; Gong et al., 2021). In essence, the utilization of pseudo labels enables the integration of unlabeled data into the learning process, thereby expanding the training set and enhancing the model's generalization capabilities. This approach effectively exploits the unlabeled data by leveraging the network's confidence in assigning pseudo labels, allowing the model to benefit from the potential knowledge contained in these un-annotated samples. Though many works (Sohn et al., 2020; Zhang et al., 2021; Wang et al., 2022d) have designed various pseudo-labeling methods and achieving noticeable improvements in the classification accuracy. We propose an orthogonal approach by formulating novel loss functions that leverage the existing powerful pseudo label generating methods.

The loss function used in semi-supervised learning is usually consisted of two parts. the (supervised) cross-entropy loss and unsupervised loss (Gong et al., 2021). Motivated by the observation that supervised cross-entropy loss has a close relationship with optimal transport (Shi et al., 2023). Considering the fact that there is a lack of in-depth understanding of modern semi-supervised

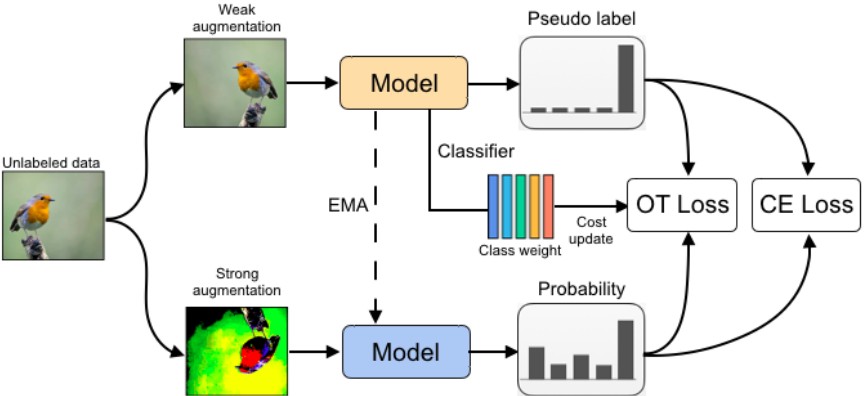

Figure 1: To obtain a pseudo-label, a model is fed with a weakly-augmented image. Then, the model predicts the probability for a strongly-augmented version of the same image. The loss includes cross-entropy and optimal transport loss, which considers the probability and pseudo-label. The cost used in optimal transport is adjusted based on the model's classification head weight.

learning, we propose to use the tool of optimal transport to give a understanding of semi-supervised learning. Additionally, when pseudo label is generated, the loss is usually given by the cross-entropy loss to make the prediction on weakly augmented image align with that of the strongly augmented counterpart. As cross-entropy loss only aligns the prediction to a specific class, leaving the relationship between classes untouched. Thus a natural question arises: Can we incorporate the information between classes to improve semi-supervised learning? We also tackle this problem by using the powerful framework of optimal transport. Specifically, we bootstrap the cost matrix used in optimal transport from the relationship between classification heads. In this manner, we successfully obtained our OTMatch framework. A diagram of OTMatch is presented in Figure 1. We then conduct the challenging experiments on various datasets with very limited labels. We observe a significant decrease in the error rate by 3.18% on CIFAR-10 with only 10 labels. This outcome underscores the simplicity and effectiveness of our approach.

Our contributions can be summarized in three folds:

- We provide an understanding of semi-supervised learning through the lens of optimal transport.
- We introduce OTMatch, a novel algorithm that exploits the inherent relationship between classess during semi-supervised learning.
- We test OTMatch on CIFAR-10/100, ImageNet, and STL-10, which consistently outperforms the state-of-the-art semi-supervised learning methods,

## 2 PRELIMINARY

### 2.1 PROBLEM SETTING AND NOTATIONS

Throughout a semi-supervised learning process, it is customary to have access to both labeled and unlabeled data. Each batch is a mixture of labeled data and unlabeled data. Assume there are $B$ labeled samples $\{(\mathbf{x}_{l_i}, \mathbf{y}_{l_i})\}_{i=1}^{B}$ and $\mu B$ unlabeled samples $\{\mathbf{x}_{u_i}\}_{i=1}^{\mu B}$ in a mixed batch, where $\mu$ is the ratio of unlabeled samples to labeled samples. We adopt the convention in semi-supervised learning (Zhang et al., 2021; Wang et al., 2022d) that there will be a teacher and student network that shares the same architecture. The teacher network does not update through gradient but by exponential moving average (EMA) instead. There will also be two sets of augmentations of different strengths, namely weak augmentation $\omega(\cdot)$ and strong augmentation $\Omega(\cdot)$.

For labeled data, the loss is the classical cross-entropy loss as follows:

$$\mathcal{L}_{\text{sup}} = \frac{1}{B} \sum_{i=1}^{B} \mathrm{H}(\mathbf{y}_{l_i}, \Pr(\omega(\mathbf{x}_{l_i}))), \tag{1}$$

where $\Pr(\omega(\mathbf{x}_{l_i}))$ denotes the output probability and $\mathrm{H}(\cdot, \cdot)$ denotes the cross-entropy loss.

The unsupervised loss $L_{un}$ is usually the main focus of improving semi-supervised learning. FixMatch (Sohn et al., 2020) introduces the idea of using a fixed threshold $\tau$ to only assign pseudo labels to those samples with enough confidence. Later, a line of works like FlexMatch (Zhang et al., 2021) and FreeMatch (Wang et al., 2022d) seeks to improve the threshold selection strategy. This loss can be formally described as follows:

$$\mathcal{L}_{\text{un1}} = \frac{1}{\mu B} \sum_{i=1}^{\mu B} \mathbf{I}(\max(\mathbf{q}_{u_i}) > \tau) \mathrm{H}(\hat{\mathbf{q}}_{u_i}, \mathbf{Q}_{u_i}), \tag{2}$$

where $\mathbf{q}_{u_i}$ is the probability of (the teacher) model on the weakly-augmented image, $\mathbf{Q}_{u_i}$ is the probability of (the student) model on the strongly-augmented image, and $\hat{\mathbf{q}}_{u_i}$ denotes the generated one-hot hard pseudo label.

FreeMatch (Wang et al., 2022d) also introduces a fairness loss to make the class distribution more balanced. The loss is given as follows:

$$\mathcal{L}_{\text{un2}} = -\mathrm{H}(\text{SumNorm}(\frac{\mathbf{p}_1}{\mathbf{h}_1}), \text{SumNorm}(\frac{\mathbf{p}_2}{\mathbf{h}_2})), \tag{3}$$

where $\mathbf{p}_1$ and $\mathbf{h}_1$ denote the mean of model predictions and histogram distribution on weakly-augmented images respectively, $\mathbf{p}_2$ and $\mathbf{h}_2$ is defined on the pseudo-labeled strongly-augmented images. As the prediction on the weakly-augmented image is more accurate, using cross-entropy loss here mimics the maximization of entropy.

## 2.2 OPTIMAL TRANSPORT

The Kantorovich formulation of discrete optimal transport (Kantorovich, 1942), also known as the transportation problem, provides a mathematical definition for finding the optimal transportation plan between two discrete probability distributions. Let's consider two discrete probability distributions, denoted as $\mu$ and $\nu$, defined on two finite sets of points, $\mathbf{X} = \{\mathbf{x}_1, \mathbf{x}_2, \ldots, \mathbf{x}_m\}$ and $\mathbf{Y} = \{\mathbf{y}_1, \mathbf{y}_2, \ldots, \mathbf{y}_n\}$, respectively. The goal is to find a transportation plan that minimizes the total transportation cost while satisfying certain constraints. The transportation plan specifies how much mass is transported from each point in $\mathbf{X}$ to each point in $\mathbf{Y}$. This is achieved by defining a transportation (plan) matrix $\mathbf{T} = [\mathbf{T}_{ij}]$, where $\mathbf{T}_{ij}$ represents the amount of mass transported from point $\mathbf{x}_i$ to point $\mathbf{y}_j$.

For ease of notation, we present the definition in the following form:

$$\min \quad \langle \mathbf{C}, \mathbf{T} \rangle$$
$$\text{subject to} \quad \mathbf{T} \in U(\mu, \nu) = \{\mathbf{T} \in \mathbb{R}_+^{m \times n} \mid \mathbf{T}\mathbf{1}_n = \mu, \mathbf{T}^T\mathbf{1}_m = \nu\},$$

where $\mathbf{C}$ denotes the cost matrix and $\langle \mathbf{C}, \mathbf{T} \rangle = \sum_{ij} \mathbf{C}_{ij} \mathbf{T}_{ij}$ is the inner product between matrices.

In this formulation, $\mathbf{c}_{ij}$ represents the cost between point $\mathbf{x}_i$ and point $\mathbf{y}_j$. It could be any non-negative cost function that captures the transportation cost between the points. The objective is to minimize the total cost, which is the sum of the products of the transportation amounts $\mathbf{T}_{ij}$ and their corresponding costs $\mathbf{c}_{ij}$. We denote the optimal transport distance as $\mathcal{W}(\mu, \nu)$.

The constraints ensure that the transportation plan satisfies the conservation of mass: the total mass transported from each point in $\mathbf{X}$ should be equal to its mass in distribution $\mu$, and the total mass received at each point in $\mathbf{Y}$ should be equal to its mass in distribution $\nu$. Additionally, the transportation amounts $\mathbf{T}_{ij}$ are non-negative.

The solution to this optimization problem provides the optimal transportation plan, which specifies how much mass is transported from each point in $\mathbf{X}$ to each point in $\mathbf{Y}$ to minimize the total cost. Algorithms, such as the Hungarian algorithm (Kuhn, 1955) can be applied to solve this problem with the complexity of $O(m^2 n)$.

The computational complexity to solve the general optimal transport problem is relatively high. Cuturi (2013) proposes an entropic regularized version of the optimal transport problem as follows:

$$\min \quad \langle \mathbf{C}, \mathbf{T} \rangle - \epsilon \mathrm{H}(\mathbf{T})$$
$$\text{subject to} \quad \mathbf{T} \in U(\mu, \nu),$$

where $\epsilon > 0$ is a hyperparameter and $\mathrm{H}(\mathbf{T}) = \sum_{i,j}(1 - \log \mathbf{T}_{ij})\mathbf{T}_{ij}$.

This regularized problem can be solved by the Sinkhorn algorithm efficiently with a complexity of $O(\frac{mn}{\epsilon})$. And it can be shown that this regularized version approximately solves the initial discrete optimal transport problem.

## 3 UNDERSTANDING FREEMATCH FROM OPTIMAL TRANSPORT PERSPECTIVE

We will use the view of optimal transport to understand one of the SOTA methods FreeMatch (Wang et al., 2022d). For simplicity, we abbreviate the EMA operation in FreeMatch.

We will first show how to use Inverse Optimal Transport (IOT) (Stuart & Wolfram, 2020; Li et al., 2019) to understand the (supervised) cross-entropy loss. Note that IOT aims to infer the cost function from the observed empirical transportation plan matrix. It usually parameterizes the cost matrix into a learnable matrix $\mathbf{C}^\theta$ and solves the following optimization problem:

$$
\begin{aligned}
\min \quad & \mathrm{KL}(\bar{\mathbf{T}} \,||\, \mathbf{T}^\theta) \\
\text{subject to} \quad & \mathbf{T}^\theta = \arg\min_{\mathbf{T} \in U(\mu,\nu)} \langle \mathbf{C}^\theta, \mathbf{T} \rangle - \epsilon \mathrm{H}(\mathbf{T}),
\end{aligned}
\tag{4}
$$

where $\bar{\mathbf{T}}$ is the (ground truth) empirical matching matrix and the definition of KL divergence is defined as follows.

**Definition 3.1.** For any two positive measures $\mathbf{P}$ and $\mathbf{Q}$ on the same support $\mathcal{X}$, the KL divergence from $\mathbf{Q}$ to $\mathbf{P}$ is given by:

$$
\mathrm{KL}(\mathbf{P} \,||\, \mathbf{Q}) = -\sum_{x \in \mathcal{X}} \mathbf{P}(x) \log \frac{\mathbf{P}(x)}{\mathbf{Q}(x)} - \sum_{x \in \mathcal{X}} \mathbf{P}(x) + \sum_{x \in \mathcal{X}} \mathbf{Q}(x).
\tag{5}
$$

The initial optimization problem is hard to solve, thus Shi et al. (2023) relax the constraints of $U(\mu, \nu)$ into $U(\mu) = \{\mathbf{T} \in \mathbb{R}_+^{m \times n} \mid \mathbf{T}\mathbf{1}_n = \mu\}$ and getting the following relaxed problem:

$$
\arg\min_{\mathbf{T} \in U(\mu)} \quad \langle \mathbf{C}, \mathbf{T} \rangle - \epsilon \mathrm{H}(\mathbf{T}).
\tag{6}
$$

When $\mu = \frac{1}{m}\mathbf{1}_m$, Shi et al. (2023) show that

$$
\mathbf{T}_{ij} = \frac{1}{m} \frac{\exp\left(-\mathbf{C}_{ij}/\epsilon\right)}{\sum_{k=1}^n \exp\left(-\mathbf{C}_{ik}/\epsilon\right)}
\tag{7}
$$

is the closed-form solution to the optimization problem (6).

Suppose the batch of labeled data is $\{(\mathbf{x}_i, \mathbf{y}_i)\}_{i=1}^B$, where $\mathbf{x}_i$ is the $i$-th image in the dataset and $\mathbf{y}_i$ is the label of this image. Assume the total number of labels is $K$.

Thus we can construct the ground truth matching matrix $\bar{\mathbf{T}}$ by setting $\bar{\mathbf{T}}_{ij} = \frac{1}{B}\delta_j^{\mathbf{y}_i}$. Denote the logits generated by the neural network for each image $\mathbf{x}_i$ as $\mathbf{l}_\theta(x_i)$. By defining the cost matrix by setting $\mathbf{C}_{ij}^\theta = c - \mathbf{l}_\theta(x_i)_j$ ($c$ is a large constant), simplifying the transport matrix (7) by dividing the same constant $\exp(-c/\epsilon)$, the transportation matrix is given by:

$$
\mathbf{T}_{ij}^\theta = \frac{1}{m} \frac{\exp(\mathbf{l}_\theta(\mathbf{x}_i)_j/\epsilon)}{\sum_{k=1}^K \exp(\mathbf{l}_\theta(\mathbf{x}_i)_k/\epsilon)}.
\tag{8}
$$

It is then straightforward to find that the loss in problem (4) is reduced as follows:

$$
\mathcal{L} = -\sum_{i=1}^B \log \frac{\exp(\mathbf{l}_\theta(x_i)_j/\epsilon)}{\sum_{k=1}^K \exp(\mathbf{l}_\theta(\mathbf{x}_i)_k/\epsilon)} + \text{Const.}
$$

Thus exactly recovers the supervised cross-entropy loss in semi-supervised learning with a temperature parameter $\epsilon$. Now we turn to understanding the more challenging unsupervised loss.

we present a lemma which will be very useful afterward in analyzing the unsupervised loss.

**Lemma 3.2.** $\frac{\sum_{i=1}^{m} s_i}{m}$ *is the unique solution of the optimization problem:*

$$\min_x \mathcal{W}(\delta_x, \sum_{i=1}^{m} \frac{1}{m} \delta_{s_i}),$$

*where the underlying cost is the square of $l^2$ distance.*

We will analyze the filtering threshold-generating protocol given by FreeMatch. Note from equation (8), we can obtain the empirical matching matrix of (the teacher) model. This threshold is generated in a self-adaptive way, namely generating the threshold based on the empirical matching matrix when lacking the ground-truth prediction confidence.

We first analyze the global threshold introduced in FreeMatch. This threshold aims to modulate the global confidence across different classes. The only information we can utilize is the matching matrix. Note each row of the matching matrix indicates the estimated probability $\mathbf{q}_{u_i}$ over the $K$ classes. An intuitive idea is to associate each unlabeled sample $u_i$ with a real number indicating the prediction confidence, thus we can take $\max(\mathbf{q}_{u_i})$ as a representative. Thus the general prediction confidence over the unlabeled data can be represented by a probability distribution $\sum_{i=1}^{\mu B} \frac{1}{\mu B} \delta_{\max(\mathbf{q}_{u_i})}$ that captures the full knowledge of the predictions. By identifying the global threshold $\tau$ with a probability distribution $\delta_\tau$ and using lemma 3.2, the global threshold can be calculated as

$$\tau = \frac{\sum_{i=1}^{\mu B} \max(\mathbf{q}_{u_i})}{\mu B}.$$

As the global threshold does not reflect the learning status of each class accurately. Then we can adjust the global threshold using each class's learning information. Note the prediction $\mathbf{q}_{u_i}$ can not only provide the "best" confidence $\max(\mathbf{q}_{u_i})$, but can also suggest the confidence on each class $k$ ($1 \le k \le K$). By collecting all the confidence of unlabeled data on class $k$ and organizing it into a probability distribution $\sum_{i=1}^{\mu B} \frac{1}{\mu B} \delta_{\mathbf{q}_{u_i}(k)}$. By using a similar argument in the global threshold case, we can get the local importance $\mathbf{p}_1(k) = \sum_{i=1}^{\mu B} \frac{\mathbf{q}_{u_i}(k)}{\mu B}$. By adjusting the relative threshold according to the importance, we can finally derive the (local) threshold as follows:

$$\tau(k) = \frac{\mathbf{p}_1(k)}{\max_{k'} \mathbf{p}_1(k')} \frac{\sum_{i=1}^{\mu B} \max(\mathbf{q}_{u_i})}{\mu B}.$$

Note when facing unlabeled data, there is no ground truth matching matrix compared to the case with labels. We thus use the prediction matrix **after** threshold filtering to serve as a "ground truth". Note both the teacher and student models share a similar format of the matching matrix given by equation (8). Thus the unlabeled samples filtered by the teacher model will also be excluded from the student model in order not to transfer predictions with little confidence. Thus the reduced matching matrices of teacher and student models are no longer probability matrices. As they still form positive matrices, by converting each row of the teacher model's predictions into pseudo labels and using the definition of KL divergence in equation (5) for positive measures. We find that the KL divergence between the teacher and student matching matrices recovers exactly the unsupervised loss $\mathcal{L}_{\text{un1}}$. The fairness loss $\mathcal{L}_{\text{un2}}$ can be understood by lemma 3.2 similarly.

**Remark**: More algorithms derived from OT framework can be found in Appendix B.

# 4  OTMATCH: IMPROVING SEMI-SUPERVISED LEARNING WITH OPTIMAL TRANSPORT

We have analyzed each component in modern semi-supervised learning in the view of optimal transport. A natural question arises: Can we improve it still using optimal transport? We answer this question in the affirmative. Note previous approaches do not explicitly consider the between-class relationships in their losses. We aim to exploit and incorporate this information into semi-supervised learning and thus improve it. Note the cost function in optimal transport plays an important role. Frogner et al. (2015) construct cost using additional knowledge like word embedding. This approach may not be problem-specific and incorporates additional knowledge. Instead, we want to bootstrap

the cost from the model itself. The basic idea is to "infer" the cost from the learning dynamic of the model. As the model parameter is updated from (the stochastic) gradient descent method, our initial step involves analyzing the gradients. To simplify the analysis, assume the feature extracted by the model as unconstrained variables. Suppose the last layer of the neural network weight is $\mathbf{W} = [\mathbf{w}_1 \mathbf{w}_2 \cdots \mathbf{w}_K]$.

Define the predicted probability matching for the image $\mathbf{x}$ as follows:

$$\mathbf{p}_k(f_\theta(\mathbf{x})) = \frac{\exp\left(f_\theta(\mathbf{x})^T \mathbf{w}_k\right)}{\sum_{k'=1}^K \exp\left(f_\theta(\mathbf{x})^T \mathbf{w}_{k'}\right)}, 1 \le k \le K.$$

Then we can calculate the loss's gradient with respect to each image embedding with a label (or pseudo label) $k$ as follows:

$$\frac{\partial \mathcal{L}}{\partial f_\theta(\mathbf{x})} = -\left(1 - \mathbf{p}_k(f_\theta(\mathbf{x}))\right) \mathbf{w}_k + \sum_{k' \ne k}^K \mathbf{p}_{k'}(f_\theta(\mathbf{x})) \mathbf{w}_{k'},$$

where $\mathcal{L}$ is the supervised loss $\mathcal{L}_{sup}$ or unsupervised loss $\mathcal{L}_{un1}$.

As the goal is to push $f_\theta(\mathbf{x})$ to the direction of $\mathbf{w}_k$, the updated score $U(\mathbf{x})$ along $\mathbf{w}_k$ when performing SGD on $f_\theta(\mathbf{x})$ can be calculated as:

$$U(\mathbf{x}) = \left\langle -\frac{\partial \mathcal{L}}{\partial f_\theta(\mathbf{x})}, \mathbf{w}_k \right\rangle = (1 - \mathbf{p}_k(f_\theta(\mathbf{x}))) \langle \mathbf{w}_k, \mathbf{w}_k \rangle - \sum_{k' \ne k}^K \mathbf{p}_{k'}(f_\theta(\mathbf{x})) \langle \mathbf{w}_{k'}, \mathbf{w}_k \rangle. \quad (9)$$

Note $U(\mathbf{x})$ reflects the hardness of classifying image $\mathbf{x}$ into class $k$. Thus we would like our expected cost of classification $C(\mathbf{x})$ to be proportional to $U(\mathbf{x})$.

Using the law of probability, we can decompose $C(\mathbf{x})$ as follows:

$$C(\mathbf{x}) = \mathbb{E}_k(\text{Cost} \mid \mathbf{x}) = \sum_{k'=1}^K \mathbf{C}_{kk'} p_{k'}(f_\theta(\mathbf{x})).$$

When $\|\mathbf{w}_i\|_2 = 1$, by taking $\mathbf{C}_{kk'} = 1 - \langle \mathbf{w}_k, \mathbf{w}_{k'} \rangle$ we can show that $C(\mathbf{x}) = U(\mathbf{x})$.

Moreover, by taking the fluctuation of batch training into consideration, we will finally derive the cost update formula as follows:

$$\mathbf{C}_{kk'} = m\mathbf{C}_{kk'} + (1 - m)(1 - \langle \mathbf{v}_k, \mathbf{v}_{k'} \rangle),$$

where the cost is initialized by discrete metric, $m$ is the momentum coefficient, and $\mathbf{v}_k = \frac{\mathbf{w}_k}{\|\mathbf{w}_k\|_2}$.

**Remark:** From the gradient update formula (9), we can get an intuitive understanding of why semi-supervised learning methods with threshold filtering work. A higher threshold makes $\mathbf{p}_k(f_\theta(\mathbf{x}))$ bigger, thus making the gradient update by this unlabeled sample "smaller". This means that even though the pseudo label may be annotated mistakenly, the effect of this mistake is relatively small by the threshold mechanism. On the other hand, the labeled data has a ground-truth gradient update. As the mistakenly annotated pseudo label is relatively small compared to the correctly annotated ones. This means that the overall update of pseudo-label-based semi-supervised learning is effective. Calculating the gradient with respect to $\mathbf{w}_k$ will result in similar arguments.

The computation cost of calculating the optimal transport cost is relatively high, which hinders its application. We present a lemma that shows under some mild conditions in semi-supervised learning, OT can be calculated in complexity $O(K)$.

**Lemma 4.1.** *Suppose two probability distributions $\mu$ and $\nu$ support on $\mathcal{X}$ and suppose $\mid \mathcal{X} \mid = K$. Suppose the cost is generated by a metric and there exists $k$ such that $\mu(i) \le \nu(i)$ for any $i \ne k$. Then $\mathcal{W}(\mu, \nu) = \sum_{i=1}^K \mathbf{C}_{ik}(\nu(i) - \mu(i))$.*

Thus we can finally obtain our optimal transport based unsupervised loss as follows:

$$\mathcal{L}_{un3} = \frac{1}{\mu B} \sum_{i=1}^{\mu B} \mathbf{I}(\max(\mathbf{q}_{u_i}) > \tau(\arg\max(\mathbf{q}_{u_i}))) \mathbf{W}(\hat{\mathbf{q}}_{u_i}, \mathbf{Q}_{u_i}).$$

Table 1: Error rates (100% - accuracy) on CIFAR-10/100, and STL-10 datasets for state-of-the-art methods in semi-supervised learning. Bold indicates the best performance, and underline indicates the second best.

| Dataset | CIFAR-10 | | | CIFAR-100 | STL-10 | |
|---|---|---|---|---|---|---|
| # Label | 10 | 40 | 250 | 400 | 40 | 1000 |
| Π Model (Rasmus et al., 2015a) | $79.18_{\pm1.11}$ | $74.34_{\pm1.76}$ | $46.24_{\pm1.29}$ | $86.96_{\pm0.80}$ | $74.31_{\pm0.85}$ | $32.78_{\pm0.40}$ |
| Pseudo Label (Lee et al., 2013) | $80.21_{\pm0.55}$ | $74.61_{\pm0.26}$ | $46.49_{\pm2.20}$ | $87.45_{\pm0.85}$ | $74.68_{\pm0.99}$ | $32.64_{\pm0.71}$ |
| VAT (Miyato et al., 2018) | $79.81_{\pm1.17}$ | $74.66_{\pm2.12}$ | $41.03_{\pm1.79}$ | $85.20_{\pm1.40}$ | $74.74_{\pm0.38}$ | $37.95_{\pm1.12}$ |
| MeanTeacher (Tarvainen & Valpola, 2017) | $76.37_{\pm0.44}$ | $70.09_{\pm1.60}$ | $37.46_{\pm3.30}$ | $81.11_{\pm1.44}$ | $71.72_{\pm1.45}$ | $33.90_{\pm1.37}$ |
| MixMatch (Berthelot et al., 2019b) | $65.76_{\pm7.06}$ | $36.19_{\pm6.48}$ | $13.63_{\pm0.59}$ | $67.59_{\pm0.66}$ | $54.93_{\pm0.96}$ | $21.70_{\pm0.68}$ |
| ReMixMatch (Berthelot et al., 2019a) | $20.77_{\pm7.48}$ | $9.88_{\pm1.03}$ | $6.30_{\pm0.05}$ | $42.75_{\pm1.05}$ | $32.12_{\pm6.24}$ | $6.74_{\pm0.17}$ |
| UDA (Xie et al., 2020) | $34.53_{\pm10.69}$ | $10.62_{\pm3.75}$ | $5.16_{\pm0.06}$ | $46.39_{\pm1.59}$ | $37.42_{\pm8.44}$ | $6.64_{\pm0.17}$ |
| FixMatch (Sohn et al., 2020) | $24.79_{\pm7.65}$ | $7.47_{\pm0.28}$ | $5.07_{\pm0.05}$ | $46.42_{\pm0.82}$ | $35.97_{\pm4.14}$ | $6.25_{\pm0.33}$ |
| Dash (Xu et al., 2021) | $27.28_{\pm14.09}$ | $8.93_{\pm3.11}$ | $5.16_{\pm0.23}$ | $44.82_{\pm0.96}$ | $34.52_{\pm4.30}$ | $6.39_{\pm0.56}$ |
| MPL (Pham et al., 2021) | $23.55_{\pm6.01}$ | $6.93_{\pm0.17}$ | $5.76_{\pm0.24}$ | $46.26_{\pm1.84}$ | $35.76_{\pm4.83}$ | $6.66_{\pm0.00}$ |
| FlexMatch (Zhang et al., 2021) | $13.85_{\pm12.04}$ | $4.97_{\pm0.06}$ | $4.98_{\pm0.09}$ | $39.94_{\pm1.62}$ | $29.15_{\pm4.16}$ | $5.77_{\pm0.18}$ |
| FreeMatch (Wang et al., 2022d) | $\underline{8.07}_{\pm4.24}$ | $4.90_{\pm0.04}$ | $4.88_{\pm0.18}$ | $37.98_{\pm0.42}$ | $15.56_{\pm0.55}$ | $\underline{5.63}_{\pm0.15}$ |
| OTMatch | $\mathbf{4.89}_{\pm7.55}$ | $\mathbf{4.72}_{\pm0.08}$ | $\mathbf{4.60}_{\pm0.15}$ | $\mathbf{37.29}_{\pm0.76}$ | $\mathbf{12.10}_{\pm0.72}$ | $\mathbf{5.60}_{\pm0.14}$ |

As FreeMatch uses $\mathcal{L}_{\text{sup}}$, $\mathcal{L}_{\text{un1}}$ and $\mathcal{L}_{\text{un2}}$. We obtain our final loss as $\mathcal{L} = \mathcal{L}_{\text{FreeMatch}} + \lambda\mathcal{L}_{\text{un3}}$, where $\lambda$ is a hyperparameter.

Interestingly, the loss $\mathcal{L}_{un3}$ can also be interpreted using the view of self-attention (Vaswani et al., 2017). Setting $f_\theta(\mathbf{x})$ as query, $\mathbf{w}_j$ ($1 \leq j \leq K$) as keys and $\mathbf{v}_j$ ($1 \leq j \leq K$) as values, recall the definition of self-attention, $\sum_{i=1}^{K} \mathbf{p}_i(f_\theta(\mathbf{x}))\mathbf{v}_i$ is exactly the representation generated by self-attention.

For an unlabeled image $\mathbf{x}$ with pseudo label $k$, the loss can be reformulated as: $\mathcal{W}(\delta_k, \Pr(\mathbf{x})) = \sum_{i=1}^{K} \mathbf{C}_{ik}\Pr(i \mid \mathbf{x}) = \sum_{i=1}^{K}(1 - \langle\mathbf{v}_i, \mathbf{v}_k\rangle)\mathbf{p}_i(f_\theta(\mathbf{x})) = 1 - \langle\sum_{i=1}^{K}\mathbf{p}_i(f_\theta(\mathbf{x}))\mathbf{v}_i, \mathbf{v}_k\rangle$. Thus intuitively, the loss $\mathcal{L}_{\text{un3}}$ wants representation generated by the self-attention mechanism to align with the classification head vector $\mathbf{v}_k$.

## 5 EXPERIMENTS

### 5.1 SETUP

To ensure a fair comparison, we follow the setup of previous studies (Sohn et al., 2020; Zhang et al., 2021; Wang et al., 2022d), conducting an evaluation of our method on widely used benchmark datasets, including CIFAR-10/100, STL-10 and ImageNet. Our algorithm is implemented based on TorchSSL, as introduced in Zhang et al.'s work (Zhang et al., 2021). It has also been extended to incorporate more advanced techniques, such as FreeMatch (Wang et al., 2022d) and SoftMatch (Chen et al., 2023). In our implementation, we integrate the optimal transport loss with the calculation of the unsupervised loss within FreeMatch. We mainly conduct experiments on the more realistic settings, i.e. the labels are limited. We employ SGD as the optimizer with a momentum of 0.9 and a weight decay of $5 \times 10^{-4}$. The learning rate is scheduled using a cosine annealing scheduler, initialized at 0.03. The batch size is configured as 64, except for on ImageNet, where it is set to 128. The ratio of unlabeled data to labeled data is set to 7. Regarding the choice of backbones, for CIFAR-10, we employ the Wide ResNet28-2. For CIFAR-100, we utilize the Wide ResNet-28-8. In the case of STL-10, we employ the Wide ResNet-37-2. Lastly, for ImageNet, we utilize the ResNet-50. We maintain the total number of training iterations at $2^{20}$ during our training process, where each step involves sampling an equal number of labeled images from all classes. For the settings of the hyperparameter of our method, we set $\lambda = 0.5$ for CIFAR-10, $\lambda = 0.15$ for STL-10 and CIFAR-100, and $\lambda = 0.01$ for ImageNet. The momentum coefficient of the cost update is set to 0.999. In all experiments, like previous work, we record the results of best performance over seeds.

### 5.2 EXPERIMENTAL RESULTS ON SEMI-SUPERVISED LEARNING

**Performance Improvements.** In our evaluation, we compare our approach to a wide range of representative semi-supervised learning methods, including Π-Model (Rasmus et al., 2015a), Pseudo-

Table 2: Error rates (100% - accuracy) on ImageNet with 100 labels per class.

|  | Top-1 | Top-5 |
|---|---|---|
| FixMatch (Sohn et al., 2020) | 43.66 | 21.80 |
| FlexMatch (Zhang et al., 2021) | 41.85 | 19.48 |
| FreeMatch (Wang et al., 2022d) | 40.57 | 18.77 |
| OTMatch | **39.29** | **17.77** |

Table 3: Ablation studies on the chosen cost in the optimal transport loss. Error rates (100% - accuracy) on CIFAR-10 with 4 labels per class are reported.

|  | Top-1 |
|---|---|
| Binary Cost | 5.20 |
| Cost Based on Covariance | 4.88 |
| OTMatch Cost | **4.72** |

Label (Lee et al., 2013), VAT (Miyato et al., 2018), MeanTeacher (Tarvainen & Valpola, 2017), MixMatch (Berthelot et al., 2019b), ReMixMatch (Berthelot et al., 2019a), UDA (Xie et al., 2020), Dash (Xu et al., 2021), MPL (Pham et al., 2021), FixMatch (Sohn et al., 2020), FlexMatch (Zhang et al., 2021), and FreeMatch (Wang et al., 2022d). The results are reported in Table 1 and 2. It is clear that OTMatch surpasses previous methods in all cases and notably improves performance, particularly on the STL-10 dataset with 40 labels and CIFAR-10 with 10 labels, compared to previous approaches. It is important to note that in CIFAR-10 cases, fully supervised has achieved an error rate of 4.62 (Wang et al., 2022d). Thus, our method further closes the gap between semi-supervised learning and fully supervised learning. Furthermore, our method can seamlessly integrate with recent and future techniques aiming at enhancing the quality of pseudo-labels, thereby achieving greater performance enhancements.

**Showcase optimal transport as a useful regularizer with minimal computation overhead.** Optimal transport can be incorporated wherever cross-entropy is used, and the computational complexity is only $O(K)$. This makes it computationally friendly, even the number of labels scales.

### 5.3 ABLATION STUDY

As the cost plays a crucial role in the optimal transport loss, we conduct further ablation studies to investigate different cost choices. A straightforward choice is the binary cost $\mathbf{c}_1(x, y) = \mathbb{I}_{x \neq y}$. However, this cost does not take into account the relationships between classes, which may lead to a decrease in performance. Additionally, we explore an alternative cost formulation that consider class relationships. In this regard, we update the cost based on the covariance matrix of predicted probabilities for strongly-augmented images.

Specifically, we compare the performance of these costs on CIFAR-10 with 40 labels benchmark and summarize the results in Table 3. It is clear that the cost used in our method achieves the best performance.

## 6 RELATED WORK

Semi-supervised learning aims to improve model performance by leveraging substantial amounts of unlabeled data and has garnered significant interest in recent years (Chen et al., 2020b; Assran et al., 2021; Wang et al., 2021). The invariance principle forms the basis for most effective semi-supervised algorithms. At its core, this principle asserts that two semantically similar images should produce similar representations when processed by the same backbone.

**Consistency regularization.** Consistency regularization, initially introduced in the Π-Model (Rasmus et al., 2015b), has emerged as a prevalent technique for implementing the invariance principle. This method has gained widespread adoption in subsequent research (Tarvainen & Valpola, 2017;

Laine & Aila, 2016; Berthelot et al., 2019b). Consistency regularization entails the generation of pseudo-labels and the application of appropriate data augmentation strategies (Tschannen et al., 2019; Berthelot et al., 2019b; Xie et al., 2020; Sohn et al., 2020; Gong et al., 2021). Pseudo-labels are created for unlabeled data and utilized in subsequent training iterations (Lee et al., 2013). The conventional approach involves minimizing the cross-entropy objective to align the predicted pseudo-labels of two distorted images, typically obtained through data augmentation (Rasmus et al., 2015b; Laine & Aila, 2016; Tarvainen & Valpola, 2017). SimMatch (Zheng et al., 2022) and CoMatch (Li et al., 2021) utilize contrastive learning to perform consistency regularization.

Moreover, extensive research has focused on generating efficient and informative pseudo-labels, taking into account various practical considerations (Hu et al., 2021; Nassar et al., 2021; Xu et al., 2021; Zhang et al., 2021; Li et al., 2022; Wang et al., 2022b). The efficacy of consistency regularization has been demonstrated as a simple yet effective approach, serving as a foundational component in numerous state-of-the-art semi-supervised learning algorithms (Sohn et al., 2020; Zhang et al., 2021).

**Improving pseudo-label quality.** In the realm of semi-supervised learning, the current discourse surrounding consistency regularization primarily revolves around augmenting the quality of pseudo-labels. Notable contributions in this domain include SimPLE, Dash, FlexMatch, CoMatch, SemCo, FreeMatch, MaxMatch, NP-Match, SEAL, and SoftMatch (Hu et al., 2021; Xu et al., 2021; Zhang et al., 2021; Li et al., 2021; Nassar et al., 2021; Wang et al., 2022d; Li et al., 2022; Wang et al., 2022a; Tan et al., 2023; Chen et al., 2023). SimPLE introduces a paired loss function that diminishes the statistical discrepancy between confident and analogous pseudo-labels, thereby enhancing their quality. Dash and FlexMatch propose dynamic and adaptable filtering techniques for pseudo-labels, which are better suited for the training process. CoMatch advocates for the integration of contrastive learning within the framework of semi-supervised learning, enabling the simultaneous learning of two representations of the training data. SemCo takes into account external label semantics to safeguard against pseudo-label quality deterioration for visually similar classes, employing a co-training approach. FreeMatch proposes a self-adjusting confidence threshold that considers the learning status of the models, allowing for improved control over pseudo-label quality. MaxMatch presents a consistency regularization technique that minimizes the most substantial inconsistency between an original unlabeled sample and its multiple augmented versions, accompanied by theoretical guarantees. NP-Match employs neural processes to amplify the quality of pseudo-labels. SEAL introduces a methodology that facilitates the concurrent learning of a data-driven label hierarchy and the execution of semi-supervised learning. SoftMatch addresses the inherent trade-off between the quantity and quality of pseudo-labels by utilizing a truncated Gaussian function to assign weights to samples based on their confidence. Taherkhani et al. (2020); Tai et al. (2021) also explore using optimal transport to improve pseudo labeling.

In this research endeavor, we adopt an unconventional approach that transcends the conventional focus on enhancing pseudo-label quality. Instead, we leverage the intrinsic relationships between classes to develop a novel methodology that integrates into existing semi-supervised learning methods, while incurring minimal computational overhead.

## 7 CONCLUSION

This paper presents a fresh perspective on addressing the challenges of semi-supervised learning, focusing on an alternative approach that goes beyond solely improving the quality of pseudo-labels. We introduce OTMatch, an innovative algorithm that harnesses the inherent relationships between classes during the training process. To effectively capture these interclass relationships, we incorporate the optimal transport distance as a loss function. By leveraging the principles of inverse optimal transport, we derive the cost between classes, allowing us to effectively align and exploit the semantic connections within the data. In our extensive experiments, we demonstrate the consistent superiority of OTMatch over state-of-the-art methods. Notably, our algorithm achieves remarkable performance while imposing minimal computational overhead. By introducing OTMatch, we not only contribute to the advancement of semi-supervised learning techniques but also provide a promising direction for future research by advocating the integration of optimal transport loss. It remains interesting to apply to self-supervised learning method like DINO (Caron et al., 2021).

## REPRODUCIBILITY STATEMENT

To foster reproducibility, we submit our experiment code as supplementary material.

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

# A MORE ON PROOFS

## A.1 PROOF OF LEMMA 3.2

*Proof.* By using the definition of Wasserstein distance. We find that $\mathcal{W}(\delta_x, \sum_{i=1}^m \frac{1}{m}\delta_{s_i}) = \frac{1}{m}\sum_{i=1}^m (x - s_i)^2$. As this is a quadratic function of $x$, we can immediately derive that the unique minimizer is $\frac{\sum_{i=1}^m s_i}{m}$. $\qquad\square$

## A.2 PROOF OF LEMMA 4.1

*Proof.* Note $\mu(i) \leq \nu(i)$ for any $i \neq k$, Thus by the probability constraints we know that $\mu(k) \geq \nu(k)$. As the cost is generated by a metric, we know that $\mathbf{C}_{kk} = 0$. Consider transporting mass from $\nu$ to $\mu$, as the cost from $k$ to $k$ is 0, $\nu$ will transport all $\nu(k)$ to $\mu(k)$. For any $i \neq k$, if $\nu$ transport $\Delta > 0$ mass to point $j$ ($j \neq k$). Then as $\nu(j) \geq \mu(j)$, $\nu$ can only transport the mass $\Delta$ to the unique point where $\nu$ has smaller mass than $\mu$. From triangular inequality $\mathbf{C}_{ik} \leq \mathbf{C}_{ij} + \mathbf{C}_{jk}$, this is costly than transporting directly from $i$ to $k$. Thus the optimal plan is to transport all the residual mass $\nu(i) - \mu(i)$ to node $k$. Thus the conclusion follows. $\qquad\square$

# B MORE ALGORITHMS DERIVED FROM OPTIMAL TRANSPORT FRAMEWORK

Shi et al. (2023) show that SimCLR (Chen et al., 2020a) and MoCo (He et al., 2019) can be understood by the optimal transport viewpoint. We would like to show that many other import algorithms can also be derived from optimal transport.

## B.1 CROSS-ENTROPY BASED CONTRASTIVE METHODS

SimMatch (Zheng et al., 2022), CoMatch (Li et al., 2021), ReSSL (Zheng et al., 2021), SwAV (Caron et al., 2020) and DINO (Caron et al., 2021) adopt the teacher student setting and use KL divergence in their loss (consider the effect of stop-gradient, the cross-entropy is equivalent to KL divergence). Similar to the derivation in Section 3, the teacher (student) matching matrices are generated by setting the cost matrix using the (negative) similarity of query samples between buffer samples (SimMatch, ReSSL), class prototypes (SwAV), other samples in a batch (CoMatch) or classification head weights (DINO). The derivation is similar to Section 3.

While both our OTMatch and contrastive learning-based methods consider the relationship between classes, there are some crucial distinctions. Our OTMatch focuses on aligning the class classification probabilities of two augmented views, following the line of work such as FixMatch, FlexMatch, and FreeMatch. In contrast, contrastive learning-based methods emphasize the consistency between two batches of augmented views. To be more specific, our OTMatch calculates each optimal transport loss exclusively involving the two augmented views. In contrast, contrastive learning-based methods such as SimMatch and CoMatch align the two batches by utilizing the representation similarity between samples in the batch. As a result, contrastive learning-based methods necessitate an additional branch, apart from the one calculating sample-wise consistency. Therefore, our OTMatch is orthogonal to contrastive learning-based methods and can be combined with them.

## B.2 CLIP

CLIP (Radford et al., 2021) is a multi-modal learning algorithm. For a batch of image text pairs $\{(I_i, T_i)\}_{i=1}^B$, the image to text loss is as follows:

$$\mathcal{L}_{I \to T} = -\sum_{i=1}^B \log \frac{\exp(\langle f_I(I_i), f_T(T_i)\rangle / \tau)}{\sum_{k=1}^B \exp(\langle f_I(I_i), f_T(T_k)\rangle / \tau)}, \tag{10}$$

where $f_I$ is the image encoder and $f_T$ is the text encoder and $\tau$ is the temperature.

The loss can be retrieved by setting the cost matrix as $\mathbf{C}_{i,j} = \|f_I(I_i) - f_T(T_j)\|_2^2$ and the ground truth matching matrix $\bar{\mathbf{T}} = diag \frac{1}{B}\mathbf{1}_B$. By noticing the fact that representations are normalized and using equation (8), calculating the loss in IOT will give the loss $\mathcal{L}_{I \to T}$. Different from uni-modal

case where there will only be transportation between the single modality. In multi-modal cases, there will also exist a symmetric transportation loss $\mathcal{L}_{T \to I}$, which can be explained by optimal transport similarly.

### B.3 SUPCON

SupCon (Khosla et al., 2020) is a supervised learning method that generates compact representations of images by incorporating label information. Suppose $I$ a batch of augmented images and $A(i) = I - \{i\}$. Denote $\mathbf{z}_i$ as the representation of image $i$, its label is $\tilde{\boldsymbol{y}}_i$.

$$\mathcal{L}_{\text{SupCon}} = \sum_{i \in I} -\frac{1}{|P(i)|} \log \sum_{p \in P(i)} \frac{\exp\left(\mathbf{z}_i \cdot \mathbf{z}_p / \tau\right)}{\sum_{a \in A(i)} \exp\left(\mathbf{z}_i \cdot \mathbf{z}_a / \tau\right)}. \tag{11}$$

Here, $P(i) \equiv \{p \in A(i) : \tilde{\boldsymbol{y}}_p = \tilde{\boldsymbol{y}}_i\}$ is the set of indices of all positives in the batch.

By setting $\mathbf{C}_{ii} = +\infty$ and $\mathbf{C}_{ij} = c - \mathbf{z}_i \cdot \mathbf{z}_j$. Noticing that the $i$-th row of ground truth matching matrix $\bar{\mathbf{T}}_{ij} = \frac{1}{I|P(i)|} \delta_{\tilde{\boldsymbol{y}}_i}^{\tilde{\boldsymbol{y}}_j}$. By noticing the fact that representations are normalized and using equation (8), calculating the loss in IOT will give the SupCon loss.

### B.4 BYOL AND SIMSIAM

BYOL (Grill et al., 2020) and SimSiam (Chen & He, 2021) uses the MSE loss between two augmented views. For a batch of images $\{\mathbf{x}_i\}_{i=1}^B$, we usually apply different augmentations to the images and get two batches of representations $\{\mathbf{z}_i^{(1)}\}_{i=1}^B$ and $\{\mathbf{z}_i^{(2)}\}_{i=1}^B$.

Take $\mu = \frac{1}{B}\mathbf{1}_B$, Shi et al. (2023) using the optimal value of the following optimization problem 12 to explain SimCLR and MoCo. We consider first change the inner minimization of the entropic regularization problem in 12 into the common optimal transport problem and get optimization problem 13. However, this bi-level optimization problem is still hard to solve. Thus we then relax the problem into a optimization problem 14.

Take the ground-truth matching matrix as $\bar{\mathbf{T}} = diag\frac{1}{B}\mathbf{1}_B$. The cost matrix $\mathbf{C}_{i,i} = \|\mathbf{z}_i^{(1)} - \mathbf{z}_i^{(2)}\|_2^2$ and $\mathbf{C}_{i,j} = c + \|\mathbf{z}_i^{(1)} - \mathbf{z}_i^{(2)}\|_2^2$ $(j \neq i)$, where $c$ is a large constant.

Then the optimization problem will be Const. $+ \frac{1}{B}\sum_i -\log \mathbf{T}_{ii} + \lambda \sum_i (\mathbf{C}_{i,i}\mathbf{T}_{i,i} + \sum_{j \neq i}(c + \mathbf{C}_{i,i})\mathbf{T}_{i,j})$. Using the constraint of $U(\mu)$ and simplifying the constant out, the objective function will be $\frac{1}{B}\sum_i(-Bc\lambda\mathbf{T}_{i,i} - \log \mathbf{T}_{i,i}) + \lambda\sum_i \mathbf{C}_{i,i} + \frac{c\lambda}{B}$. As $\mathbf{T} \in U(\mu)$, the optimal value is Const. $+ \lambda\sum_i \mathbf{C}_{ii}$. This exactly recovers the MSE loss.

$$
\begin{aligned}
\min \quad & \text{KL}(\bar{\mathbf{T}} \,\|\, \mathbf{T}^\theta) \\
\text{subject to} \quad & \mathbf{T}^\theta = \underset{\mathbf{T} \in U(\mu)}{\arg\min} \langle \mathbf{C}^\theta, \mathbf{T} \rangle - \epsilon \text{H}(\mathbf{T}).
\end{aligned}
\tag{12}
$$

$$
\begin{aligned}
\min \quad & \text{KL}(\bar{\mathbf{T}} \,\|\, \mathbf{T}^\theta) \\
\text{subject to} \quad & \mathbf{T}^\theta = \underset{\mathbf{T} \in U(\mu)}{\arg\min} \langle \mathbf{C}^\theta, \mathbf{T} \rangle.
\end{aligned}
\tag{13}
$$

$$
\begin{aligned}
\min \quad & \text{KL}(\bar{\mathbf{T}} \,\|\, \mathbf{T}) + \lambda \langle \mathbf{C}, \mathbf{T} \rangle \\
\text{subject to} \quad & \mathbf{T} \in U(\mu).
\end{aligned}
\tag{14}
$$

### B.5 MORE DISCUSSIONS ON DIFFERENCES WITH (TAI ET AL., 2021)

While OT was not the first time being used in semi-supervised learning, there are significant differences between our method and previous work (Tai et al., 2021), as we pointed out in the related work section. We would like to provide a more detailed discussion as follows. First, we leverage OT theory to improve the loss function, differing from (Tai et al., 2021), which utilizes OT for refining pseudo-label assignments, reflecting distinct objectives. Second, our method has computational efficiency and ease of implementation, in contrast to (Tai et al., 2021), which introduces computational burdens. Third, (Tai et al., 2021) lags behind in accuracy compared to modern threshold-based pseudo-labeling strategies, whereas our method achieves SOTA results on benchmarks among threshold-based pseudo-labeling methods.

## C IMPLEMENTATION ALGORITHM

We will present any algorithm sketch for our OTMatch method. It is very clear that our method only needs a few more lines compared to FreeMatch.

---

**Algorithm 1** OTMatch training algorithm at $t$-th step

---

1: **Input:** Number of classes $K$, labeled samples $\{(\mathbf{x}_{l_i}, \mathbf{y}_{l_i})\}_{i=1}^{B}$, unlabeled samples $\{\mathbf{x}_{u_i}\}_{i=1}^{\mu B}$, FreeMatch loss weights $w_1$, $w_2$, and EMA decay $m$, OT loss balancing weight $\lambda$, normalized classification head vectors $\{\mathbf{v}_i\}_{i=1}^{K}$.
2: **FreeMatch loss:**
3: Calculate $\mathcal{L}_{\text{sup}}$ using equation (1)
4: $\tau_t = m\tau_{t-1} + (1-m)\frac{1}{\mu B}\sum_{i=1}^{\mu B}\max(\mathbf{q}_{u_i})$
5: $\tilde{p}_t = m\tilde{p}_{t-1} + (1-m)\frac{1}{\mu B}\sum_{b=1}^{\mu B}\mathbf{q}_{u_i}$
6: $\tilde{h}_t = m\tilde{h}_{t-1} + (1-m)\,\text{Hist}_{\mu B}(\hat{\mathbf{q}}_{u_i})$
7: **for** $c = 1$ to $K$ **do**
8: $\quad \tau_t(c) = \text{MaxNorm}(\tilde{p}_t(c)) \cdot \tau_t$
9: **end for**
10: Calculate $\mathcal{L}_{\text{un1}}$ using equation (2)
11: $\overline{p} = \frac{1}{\mu B}\sum_{i=1}^{\mu B}\mathbf{I}\left(\max(\mathbf{q}_{u_i}) \geq \tau_t(\arg\max(\mathbf{q}_{u_i}))\right)\mathbf{Q}_{u_i}$
12: $\overline{h} = \text{Hist}_{\mu B}(\mathbf{I}\left(\max(\mathbf{q}_{u_i}) \geq \tau_t(\arg\max(\mathbf{q}_{u_i}))\hat{\mathbf{Q}}_{u_i}\right))$
13: Calculate $\mathcal{L}_{\text{un2}}$ using equation (3)
14: $\mathcal{L}_{\text{FreeMatch}} = \mathcal{L}_{\text{sup}} + w_1\mathcal{L}_{\text{un1}} + w_2\mathcal{L}_{\text{un2}}$
15: **Update cost:**
$\quad \mathbf{C}_t(i,j) = m\mathbf{C}_{t-1}(i,j) + (1-m)(1 - \langle \mathbf{v}_i, \mathbf{v}_j \rangle)$
16: **OT loss:**
$\quad \mathcal{L}_{\text{un3}} = \frac{1}{\mu B}\sum_{i=1}^{\mu B}\mathbf{I}(\max(\mathbf{q}_{u_i}) > \tau_t(\arg\max(\mathbf{q}_{u_i})))\sum_{k=1}^{K}\mathbf{C}_t(\arg\max(\mathbf{q}_{u_i}), k)\mathbf{Q}_{u_i}(k)$
17: **OTMatch loss:**
$\quad \mathcal{L} = \mathcal{L}_{\text{FreeMatch}} + \lambda\mathcal{L}_{un3}$

---

## D FURTHER ANALYSIS

### D.1 RESULTS ON NLP TASKS.

To demonstrate the utility of our approach, we further extend our evaluations to encompass USB datasets (Wang et al., 2022c) of language modality. Specifically, the results in Table 4 demonstrate that on both Amazon Review and Yelp Review, when our approach is integrated with Flex-Match (current state-of-the-art), we have achieved an improvement, reaching a new state-of-the-art. This also validates the fact that, beyond the compatibility with FreeMatch, our approach actually can be effortlessly integrated with various existing methods.

Table 4: Comparisons with state-of-the-art semi-supervised learning methods on Amazon Review and Yelp Review. Error rates (100% - accuracy) are reported.

| | Amazon Review | | Yelp Review | |
|---|---|---|---|---|
| **Method** | 250 labels | 1000 labels | 250 labels | 1000 labels |
| FixMatch | 47.85 | 43.73 | 50.34 | 41.99 |
| CoMatch | 48.98 | 44.37 | 46.49 | 41.11 |
| CRMatch | 46.23 | 43.69 | 46.61 | 41.80 |
| AdaMatch | 46.75 | 43.50 | 48.16 | 41.71 |
| SimMatch | 47.27 | 43.09 | 46.40 | 41.24 |
| FlexMatch | 45.75 | 43.14 | 46.37 | 40.86 |
| FlexMatch + OT | **43.81** | **42.35** | **43.61** | **39.76** |

Table 5: Ablation study of the loss weight on CIFAR-10 40 labels

| $\lambda$ | 0.3 | 0.4 | 0.5 | 0.6 | 0.7 |
|---|---|---|---|---|---|
| **Error rate** | 4.91 | 4.95 | 4.72 | 4.97 | 4.98 |

## D.2 ABLATIONS OF THE LOSS WEIGHT

We conduct additional experiments to assess how the weight of the optimal transport loss impacts performance. The results are presented in Table 5, where the loss weight is denoted by $\lambda$. It can be observed that when lambda is set to 0.5, our method achieves the best performance, and the results are not sensitive to changes in the loss weight when the variation range is small.

## D.3 ANALYSIS OF THE RUNNING TIME

We calculate the per-iteration running time of FreeMatch and OTMatch on CIFAR-10 with 40 labels. From Table 6, it can be observed that our method introduces a small computation overhead.

Table 6: Per-iteration running time

| | |
|---|---|
| FreeMatch | 0.23 s |
| OTMatch | 0.25 s |

