# OpenReview forum: "OTMatch: Improving Semi-Supervised Learning with Optimal Transport"
_ICLR.cc/2024/Conference — Submitted to ICLR 2024_

### Official Review · Reviewer_RN81 · 2023-10-29

**Soundness:** 2 fair
**Presentation:** 2 fair
**Contribution:** 2 fair
**Rating:** 3
**Confidence:** 5

**Summary:**

This paper discusses the limitation of cross-entropy loss in addressing relationships between classes and introduces the OTMatch framework, which uses the optimal transport method to improve semi-supervised learning by incorporating inter-class information. The proposed method is evaluated on CIFAR-10/100, ImageNet, and STL-10.

**Strengths:**

1. The motivation behind this work is evident.
2. The overall organization the paper is commendable, making it easy to follow and understand.

**Weaknesses:**

1. The novelty is limited. Using OT strategy to address semi-supervised learning has been proposed in previous work [1]. The proposed method does not show significant improvement compared to existing work and lacks the necessary experimental comparisons.
2. The improvement of proposed method on the evaluated dataset is not significant, and there is no comparison regarding its impact on convergence speed and the loss curve.

[1] K. S. Tai, P. D. Bailis, and G. Valiant, “Sinkhorn label allocation: Semi-supervised classification via annealed self-training,” in International
Conference on Machine Learning. PMLR, 2021

**Questions:**

It is suggested to add more analysis about the impact of long-tail classes on the OT approach, and more analysis of the advantage/disadvantage of the threshold-free methods.

---

> ### Author Response · Authors · 2023-11-18
> **Response to reviewer RN81**
>
> >Q1: The novelty is limited. Using OT strategy to address semi-supervised learning has been proposed in previous work [1]. The proposed method does not show significant improvement compared to existing work and lacks the necessary experimental comparisons.
>
> A1: Thank you for your comments. **While OT was not the first time being used in semi-supervised learning, there are significant differences between our method and previous work [1], as we pointed out in the related work section in the initial manuscript.** We would like to provide a more detailed discussion as follows. First, we leverage OT theory to improve the loss function, differing from [1], which utilizes OT for refining pseudo-label assignments, reflecting distinct objectives. Second, our method has computational efficiency and ease of implementation, in contrast to [1], which introduces computational burdens. Third, [1] lags behind in accuracy compared to modern threshold-based pseudo-labeling strategies, whereas our method achieves SOTA results on benchmarks among threshold-based pseudo-labeling methods. These distinctions underscore the novelty and superiority of our approach, and we have incorporated the above discussion into the revised version of our paper.
>
> >Q2: The improvement of proposed method on the evaluated dataset is not significant, and there is no comparison regarding its impact on convergence speed and the loss curve.
>
> A2: Thank you for your comments. **We would like to emphasize that the performance improvement achieved by combining our OT loss with the pseudo-labeling approach in semi-supervised learning is absolutely noteworthy.** This is particularly evident in scenarios where the number of labeled samples is extremely limited. Compared to the state-of-the-art method FreeMatch, we achieve a 3.18% and 3.46% reduction in error rate on CIFAR-10 with 1 label per class and STL-10 with 4 labels per class. Our improvement is achieved solely through a simple optimal transport loss, with almost no additional training cost introduced.
>
> We also provide training details about OTMatch to enhance understanding. The core of our method lies in learning a cost matrix. Therefore, at the initial stages of training, OTMatch achieves performance similar to FreeMatch since the cost matrix initialized may lack prior information about classes. As training progresses and the cost matrix continuously updates, typically in the later stages around the predefined 1.04 million iterations, OTMatch begins to show the potential to outperform FreeMatch.
> We have provided a detailed showcase of how the optimal performance of FreeMatch and OTMatch evolves on ImageNet after one million iterations, as follows.
> |  Iterations | 978000  |1011000|1013000|  1027000  | 1043000  |
> |  - | -  |  -  | -  | -  | ----  |
> | FreeMatch Best Accuracy  |0.5795|0.58104|0.58364|0.59026|0.59428|
>
> |  Iterations | 976000  |1007000|1013000|1022000| 1030000|1035000 | 1038000  |1040000  |
> |  -  | -  |  -  | ----  | ----  | ----  | ----  | ----  | ----  |
> | OTMatch Best Accuracy|0.57792|0.58364|0.58714|0.59262|0.59938|0.60188|0.60484|0.6071|
>
> It can be observed that OTMatch achieves performance comparable to FreeMatch before one million iterations but gradually surpasses FreeMatch later.
>
> Additionally, we conduct a more in-depth investigation, specifically examining the impact of discarding the cost matrix midway through training. In particular, we experiment with training OTMatch on ImageNet for one million iterations, followed by removing the OTLoss directly or resetting the cost matrix to ones. Then we observe how their performance changes after the next 1000 iterations. The results are shown below.
>
> |  | original  |remove|reset|
> |  -  | -  |  -  |  -  |
> | Accuracy|0.57832|0.56125|0.54927|
>
> It is evident that discarding or resetting a learned cost matrix significantly affects performance. This demonstrates that the cost matrix has indeed learned crucial information about inter-class relationships, highlighting the effectiveness of our approach.
>
> >Q3: It is suggested to add more analysis about the impact of long-tail classes on the OT approach, and more analysis of the advantage/disadvantage of the threshold-free methods.
>
> A3: Thank you for your suggestion. Following the conventional semi-supervised learning works, we focus on the standard settings of semi-supervised learning in this paper. Regarding the long-tail considerations are outside the scope of this paper, and we would like to leave them as potential future work.
> However, we believe our method will be beneficial under long-tail cases because the semantic relationship between classes may help improve classification. The threshold-free methods have the advantages of clearer optimization-based pseudo-label assignment strategies and no heavy hyperparameter tuning. **They have the disadvantages of much longer running times and lower empirical accuracies compared to modern threshold-based pseudo-labeling methods.**

---

> > ### Comment · Reviewer_RN81 · 2023-11-21
> >
> > Thanks for the exhaustive explanations. However, there are concerns over completeness of paper and the empirical results. After reading all reviews and replys, so I'd like to keep my score.

---

> > > ### Author Response · Authors · 2023-11-21
> > > **Kindly request for providing specific questions for further discussion**
> > >
> > > Dear Reviewer RN81,
> > >
> > > Thank you for taking the time to review our paper and for considering our rebuttal. We appreciate your acknowledgment of our exhaustive explanations. To address your reserved concerns about our paper, we would like to **highlight some key points** in our rebuttal.
> > >
> > > 1. Regarding your concern about novelty, we have emphasized **three crucial distinctions** between our method and related work [1], underscoring the novelty and superiority of our approach. The citation for [1] has already been included, and the discussion has been integrated into the revised paper.
> > >
> > > 2. Regarding your concern about empirical performance, we would like to reiterate that, when compared to the state-of-the-art method FreeMatch, **we achieve a 3.18% and 3.46% reduction in error rate on CIFAR-10 with 1 label per class and STL-10 with 4 labels per class**, respectively. This improvement is achieved solely through a simple optimal transport loss, with almost no additional training cost introduced. Moreover, **our method demonstrates a notable 1.28% enhancement in top-1 accuracy on the larger dataset ImageNet** over the SOTA method FreeMatch -- a comparable improvement to FreeMatch's improvement over the previous SOTA method FlexMatch. **It is worth noting that these gains on ImageNet are significant not only due to the dataset's scale but also because our method is both swift and straightforward to implement.**
> > >
> > > Despite the points mentioned above, we acknowledge your persistent concerns regarding the paper's completeness and the empirical results. In order to better address these concerns, **we kindly request that you provide specific details, questions, or suggestions for further discussion.** Your input would be immensely valuable in refining our work and addressing any outstanding issues.

---

> ### Author Response · Authors · 2023-11-20
> **We would be grateful if you could take a look at the response**
>
> Dear Reviewer RN81:
>
> We sincerely appreciate your valuable time devoted to reviewing our manuscript. We would like to gently remind you of the approaching deadline for the discussion phase. We have diligently addressed the issues you raised in your feedback, providing detailed explanations. As an example, we have made more discussions of the distinctions of our method and the prior work Sinkhorn label allocation. We also add these new discussions into our revised manuscript. Would you kindly take a moment to look at it?
>
> We are very enthusiastic about engaging in more in-depth discussions with you.

---

> ### Author Response · Authors · 2023-11-22
> **Seeking Your Input on Revised Paper's Alignment with ICLR Standards**
>
> Dear Reviewer RN81,
>
> As the discussion period approaches its conclusion, **we want to ensure that we have thoroughly addressed all your concerns and that our revised paper fully meets the standards of ICLR**. We would highly value any additional feedback you may provide.
>
> Thank you sincerely for your time and consideration.
>
> Best regards,
>
> The Authors

---

### Official Review · Reviewer_gfig · 2023-10-31

**Soundness:** 3 good
**Presentation:** 3 good
**Contribution:** 2 fair
**Rating:** 3
**Confidence:** 4

**Summary:**

This paper proposes a method that leverages the inherent relationships among existing classes by using optimal transport.

**Strengths:**

1. The paper proposes an interesting idea that exploits the inherent relationship between classes in semi-supervised learning

**Weaknesses:**

1. Ablation study on \(\epsilon\) is missing? How did you determine the appropriate \(\epsilon\) value for OT?
2. The novelty is limited. In comparison to Freematch, this paper introduces an additional loss, \( L_{un3} \). However, an ablation study detailing \( L_{un3} \) under various hyperparameters seems to be absent.
3. Section 4 lacks clarity and could benefit from further elucidation.
4. I strongly recommend presenting a structured algorithm or providing a comprehensively defined overall loss function for better clarity.

**Questions:**

1. In the text following the second formulation, "q_ui" should be corrected to "Q_ui."
2. The depiction in Figure 1 is ambiguous. What constitutes the input for the OT loss?
3. How can we obtain the ground truth matching matrix \( T \) in formulation (1)?
4. Why was KL divergence chosen as the loss function?
5. I am unclear about the statement in the paper that suggests the objective is "to push \( f(x) \) in the direction of \( wk \)." Can this be clarified?

---

> ### Author Response · Authors · 2023-11-18
> **Response to reviewer gfig**
>
> >Q1: Ablation study on (\epsilon) is missing? How did you determine the appropriate (\epsilon) value for OT?
>
> A1: Thank you for your question. We want to clarify that we do not use the entropic regularized method to compute OT. Instead, for semi-supervised learning, the initial (unregularized) OT loss has a closed-form solution which has O(K) running time faster than the Sinkhorn algorithm for entropic regularized OT. **Thus, we employ the closed form in our method and do not need to determine $\epsilon$.**
>
> > Q2: The novelty is limited. In comparison to Freematch, this paper introduces an additional loss, ( L_{un3} ). However, an ablation study detailing ( L_{un3} ) under various hyperparameters seems to be absent.
>
> A2: Thank you for your comments. **We would like to highlight that we first understand FreeMatch through an optimal transport lens and then improve it using an additional OT loss. This makes our method a unified one from an optimal transport view.** In addition, our method is easy to implement, fast, and effective.
> In our experiments, we initially set the loss weight to 0.5. In response to your inquiry about the ablation study, we conducted additional experiments focusing on evaluating the impact of different loss weights on the model's performance. The corresponding results are presented in the table below, where the loss weight is represented by the variable lambda.
> | $\lambda$ | 0.3 | 0.4 | 0.5 | 0.6 | 0.7 |
> | --- | --- | --- | --- | --- | --- |
> | Error rate | 4.91 | 4.95 | 4.72 | 4.97 | 4.98 |
>
> Based on the table, we can observe that our method achieves the optimal performance when lambda is set to 0.5. In comparison, other related works like FreeMatch exhibit greater sensitivity to the choice of loss weight, emphasizing its impact on the final performance. We have included this ablation study in the revised version of our paper for reference.
>
> >Q3: Section 4 lacks clarity and could benefit from further elucidation.
>
> A3: Thank you for your comments. **The section mainly discusses how to choose the cost matrix naturally from gradient analysis and provides a closed from and fast solution to optimal transport.** In addition, we have added the algorithm in Appendix C to make it clearer.
>
> >Q4: I strongly recommend presenting a structured algorithm or providing a comprehensively defined overall loss function for better clarity.
>
> A4: **Thank you for your suggestion and we have added an algorithm box to our revised paper in Appendix C.**
>
> >Q5: In the text following the second formulation, "q_ui" should be corrected to "Q_ui."
>
> A5: Thank you for pointing out the typo. **We have corrected it in our revised paper.**
>
> >Q6: The depiction in Figure 1 is ambiguous. What constitutes the input for the OT loss?
>
> A6: Thank you for your feedback. As we depicted in the initial manuscript, the input probabilities are the one hot pseudo-label from a weakly augmented image and the probability prediction of a strongly augmented image. **As OT needs a cost, we use the classification head vectors to calculate the cost.** You can see more clearly from our algorithm.
>
> >Q7: How can we obtain the ground truth matching matrix ( T ) in formulation (1)?
>
> A7: Thank you for your feedback. **As we depicted in the initial manuscript, when we tackle labeled samples, we can organize the labeling function as an empirical matching matrix where each row depicts a labeled sample and the only non-zero column in this row is its true label.**
>
> >Q8: Why was KL divergence chosen as the loss function?
>
> A8: Thank you for your feedback. The reasons are two-fold. One is the empirical matching matrices and the ground truth matrix can be seen as a probability distribution, KL divergence is a typical way to measure differences between probabilities. **The other is that we adopt an (inverse) optimal transport way to understand FreeMatch, where in the definition of inverse optimal transport, it adopts KL divergence.**
>
> >Q9: I am unclear about the statement in the paper that suggests the objective is "to push ( f(x) ) in the direction of ( wk )." Can this be clarified?
>
> A9: Thank you for your feedback. Recall that in calculating logits from representation, logits are the inner product of representation with each classification head. **If the sample is from class k, a very confident classification will be logit k is far larger than other logits.** As logit k is the inner product of representation $f(x)$ and $w_k$, making this inner product larger geometrically means that $f(x)$ is closer to the direction of $w_k$.

---

> ### Author Response · Authors · 2023-11-20
> **We would be grateful if you could take a look at the response**
>
> Dear Reviewer gfig:
>
> We sincerely appreciate your valuable time devoted to reviewing our manuscript. We would like to gently remind you of the approaching deadline for the discussion phase. We have diligently addressed the issues you raised in your feedback, providing detailed explanations. As an example, we have included an additional algorithm block in Appendix C of the revised manuscript as you required. This addition aims to provide a clearer and more detailed explanation of the simplicity of our new OT loss. Would you kindly take a moment to look at it?
>
> We are very enthusiastic about engaging in more in-depth discussions with you.

---

> ### Comment · Reviewer_gfig · 2023-11-20
> **Reply to author**
>
> Thank you for your answer, but based on your response, I still have questions.
>
> 1. If the ground truth matching matrix T only considers labeled data, does L3 also only consider the labeled set? This seems unnecessary. If not, how is the ground truth matching matrix for the unlabeled set obtained?
>
> 2. You mention that "our method is a unified one from an optimal transport view."  However, to me, your method appears similar to [1]. Could you point out the differences and the novelties in your approach?
>
> 3. For the additional ablation study, why not include λ=0? Were the experiments conducted multiple times? The results are very close, which does not seem convincing
>
> [1] Understanding and Generalizing Contrastive Learning from the Inverse Optimal Transport Perspective, ICML 2023

---

> ### Author Response · Authors · 2023-11-21
> **Further Response to Reviewer gfig**
>
> Dear Reviewer gfig:
>
> We sincerely appreciate your effort in reviewing our rebuttal and providing further feedback. In response to your comments, we would like to offer the following additional clarifications.
>
> ---
>
> > Q1: If the ground truth matching matrix T only considers labeled data, does L3 also only consider the labeled set? This seems unnecessary. If not, how is the ground truth matching matrix for the unlabeled set obtained?
>
> A1: Thank you for your follow-up question. We want to clarify that the loss $L_{un3}$ **does not take the labeled data into account**, and it only handles the unlabeled data.
>
> Regarding how to obtain the ground-truth matching matrix, we would like to provide more details to complement our previous answer. As we discussed in the last paragraph of section 3, the case of the unlabeled set does not have a real ground-truth matching matrix in the supervised case. Instead, **we use the matching matrix of the teacher model after threshold filtering** (discard the rows in the teacher matching matrix that this row does not have a pseudo-label after threshold filtering) to serve as a "ground truth".
>
> > Q2: You mention that "our method is a unified one from an optimal transport view." However, to me, your method appears similar to [1]. Could you point out the differences and the novelties in your approach?
>
> A2: Thank you for your question. First of all, we would like to highlight that **our main focus is to demonstrate that the FreeMatch algorithm can be naturally understood using optimal transport (OT) for each individual loss**. This makes our OTMatch algorithm easily interpretable within the OT framework. Although OTMatch may seem to simply add an OT loss on top of FreeMatch, the understanding of OT within FreeMatch allows OTMatch to be a well-interpretable algorithm within the OT framework.
>
> Regarding your question, let us further explain the differences between our approach and the one presented in [1]. Firstly, in the semi-supervised setting, the $L_{un}$ term mainly arises from the loss generated by teacher-student architecture, which is different from the fully supervised scenario studied in [1]. Secondly, we provide an OT interpretation for the selection of thresholds, which is a unique operation in the semi-supervised context.
>
> > Q3: For the additional ablation study, why not include λ=0? Were the experiments conducted multiple times? The results are very close, which does not seem convincing.
>
> A3: Thank you for your comments. Our initial motivation for this ablation study was to assess the impact of varying $\lambda$ around the value we used in our experiments ($\lambda=0.5$) to **determine whether 0.5 is the optimal value**. Thus, we did not include 0 in our previous rebuttal, which is far from the selected value of 0.5. In response to your question, **we conduct additional experiments on** $\lambda=0$ and observe that it underperforms the other $\lambda$ values. (ps. Our re-implementation with a value of 0 is a little bit worse than the reported 4.90 in the original paper, but it still can achieve better performance with the optimal value of $\lambda=0.5$.)
>
> | $\lambda$ | 0 | 0.3 | 0.4 | 0.5 | 0.6 | 0.7 |
> | --- | --- | --- | --- | --- | --- | --- |
> | Error rate | 5.21 | 4.91 | 4.95 | **4.72** | 4.97 | 4.98 |
>
> Secondly, we did not perform multiple runs in this ablation study as we followed the conventional approach adopted in previous related works, which typically involves conducting a single run. Moreover, we refer the reviewer to **our main experiments of the paper which are conducted in multiple runs**, adequately demonstrating the effectiveness of our method.
>
> Thirdly, the relatively small changes observed are attributed to the high accuracy achieved on the simple CIFAR-10 dataset, making the differences less pronounced. **As a reference, the difference in performance between FreeMatch and the previous SOTA FlexMatch is only 0.07.**
>
> ---
>
> We hope this response addresses your concerns, and we are **open to further discussion** to ensure the quality and clarity of our work.

---

> > ### Comment · Reviewer_gfig · 2023-11-21
> > **Reply to author**
> >
> > Thanks for your response. However, compared to [1], which explains contrastive learning by IoT, your method simply applies the same concept, but it sets the matching matrix corresponding to values greater than the threshold to 1, which means the contribution is not enough. Therefore, I'd like to keep my score.
> >
> > [1] Understanding and Generalizing Contrastive Learning from the Inverse Optimal Transport Perspective, ICML 2023

---

> ### Author Response · Authors · 2023-11-21
> **Further Response to Reviewer gfig**
>
> Dear Reviewer gfig,
>
> We sincerely appreciate your valuable time devoted to reviewing our rebuttal and providing further comments.
>
> First and foremost, we would like to reiterate the main contribution of our paper. While [1] explains "supervised learning" and contrastive "self-supervised learning" using OT, there lacks an explanation of the important paradigm "semi-supervised learning" using OT. Thus, our motivation is to **resolve the last piece of the puzzle**.
>
> However, it is not trivial to adopt [1]'s idea for semi-supervised learning. This is because **semi-supervised learning is a more challenging problem** than supervised learning considered by [1]. It involves new issues such as 1) how to adaptively determine the threshold, and 2) how to obtain pseudo-labels based on the threshold. Our proposed method addresses the above two issues and achieves SOTA performance on various semi-supervised benchmarks.
>
> Secondly, we want to emphasize again that providing an OT interpretation of FreeMatch has more implications. It facilitates incorporating an additional OT loss into it naturally, making OTMatch a unified framework for semi-supervised learning. **Without an OT explanation for FreeMatch, it is hard to see OTMatch as a unified algorithm based on OT.**
>
> Thirdly, despite the theoretical aspects, the proposed method also has empirical significance, achieving SOTA performance. In particular, when compared to the SOTA method FreeMatch, we **achieve a 3.18% and 3.46% reduction in error rate** on CIFAR-10 with 1 label per class and STL-10 with 4 labels per class, respectively. This improvement is achieved solely through a simple optimal transport loss, with almost no additional training cost introduced. Moreover, our method demonstrates a **notable 1.28% enhancement in top-1 accuracy on the larger dataset ImageNet** over SOTA method FreeMatch -- a comparable improvement to FreeMatch's improvement over the previous SOTA method FlexMatch. **It is worth noting that these gains on ImageNet are significant not only due to the dataset's scale but also because our method is both swift and straightforward to implement.**
>
> Based on the above justifications, our work demonstrates the strong power of OT. It not only **provides a new understanding of SOTA method in semi-supervised learning**, but also directly **spawns a new approach in semi-supervised learning**.
>
> ---
>
> Having **addressed the majority of your concerns** in the initial review (Q1-Q9) and follow-up questions (Q1-Q3), and having provided additional elaboration on the single remaining issue in this response, **we respectfully request a reconsideration of the scoring**. We firmly believe that our research offers valuable insights and contributions that would greatly benefit the ICLR community.

---

> ### Author Response · Authors · 2023-11-22
> **Seeking Your Input on Revised Paper's Alignment with ICLR Standards**
>
> Dear Reviewer gfig,
>
> As the discussion period approaches its conclusion, **we want to ensure that we have thoroughly addressed all your concerns and that our revised paper fully meets the standards of ICLR**. We would highly value any additional feedback you may provide.
>
> Thank you sincerely for your time and consideration.
>
> Best regards,
>
> The Authors

---

### Official Review · Reviewer_29Ss · 2023-10-31

**Soundness:** 3 good
**Presentation:** 3 good
**Contribution:** 3 good
**Rating:** 5
**Confidence:** 5

**Summary:**

This paper focuses on semi-supervised learning and proposes a new algorithm called OTMatch. The proposal aims to capture relationships between classes and adopt the optimal transport distance as a loss function. Experimental results show that the proposal can achieve performance improvement over previous SSL methods on some benchmark datasets.

**Strengths:**

This paper proposed a new SSL algorithm, compared with previous SSL methods, the proposal considers capturing the relationship between classes. The idea is insightful to some extent.

**Weaknesses:**

1. Why the class relationship is helpful for semi-supervised learning? Is there any analysis or discussion?
2. If the class relationship is important for semi-supervised learning, is there any easier method to exploit the relationship?
3. The proposed unsupervised loss is difficult to compute. Although the authors claim in some conditions, the computational complexity can be reduced, whether these conditions are satisfied in real-world tasks is hard to know.
4. When there are many classes (such as ImageNet or CIFAR-100), the performance improvement is limited.

**Questions:**

As discussed above.

---

> ### Author Response · Authors · 2023-11-18
> **Response 1 to reviewer 29Ss**
>
> >Q1: Why is the class relationship helpful for semi-supervised learning? Is there any analysis or discussion?
>
> A1: Thank you for your question. Semi-supervised learning usually has very few labelled samples compared to the unlabeled ones. When there are only very limited annotated data, the predicted pseudo-label will be less reliable. More importantly, the pseudo-label method will cause a “overconfidence” issue [r1], which means the model will usually fit on the confident but wrong pseudo-labels, resulting in poor performance. In pseudo-label based semi-supervised learning, cross-entropy loss is usually used, but it only considers pushing the prediction to their respective pseudo-labels and does not take the relationship between classes into account. Thus using cross-entropy loss alone will make the model fit on wrong label. **By introducing optimal transport loss that takes the class-relationship into account, we expect it to overcome this over-confidence drawback of cross-entropy loss in semi-supervised learning.** Intuitively, even when the pseudo label is wrong, as the wrong classified classes usually have close semantic relationship with the correct class, introducing an optimal transport loss will use this semantic relationship to update the cost and may help mitigate the over-confidence effect.
>
> [r1]: Semi-Supervised Learning with Multi-Head Co-Training, Mingcai Chen, Yuntao Du, Yi Zhang, Shuwei Qian, Chongjun Wang, AAAI 2022
>
> >Q2: If the class relationship is important for semi-supervised learning, is there any easier method to exploit the relationship?
>
> A2:  Thank you for your question. **We consider a method "easy" based on its simplicity, involving small complexity (both in terms of time and space), as well as ease of implementation. Regarding time complexity, our optimal transport loss achieves O(K), representing a relatively low computational cost.** This ensures the efficiency of our method theoretically, a feature lacking in some related works such as FreeMatch [r2]. Moreover, we conduct additional experiments in Appendix D.3 on real running time and observe that our method indeed has a very close runtime to previous related works. (i.e., the running time increases by 0.02s compared to FreeMatch during every iteration). Concerning space complexity, our method introduces only a constant additional memory overhead, resulting in O(1) space complexity. In terms of implementation, our method leverages semantic information among classes with just one line of code (see Appendix C), making it straightforward to implement. In conclusion, our method is designed to be sufficiently easy to effectively exploit class relationships in semi-supervised learning.
>
> [r2] FreeMatch: Self-adaptive thresholding for semi-supervised learning.
>
> >Q3: The proposed unsupervised loss is difficult to compute. Although the authors claim in some conditions, the computational complexity can be reduced, whether these conditions are satisfied in real-world tasks is hard to know.
>
> A3: Thank you for your feedback. **It's important to clarify that, in all our experiments where pseudo-labels are one-hot vectors, the O(K) time complexity of our proposed method is strictly satisfied without any additional conditions.** This is supported by lemma 4.1, the condition is met when one probability vector has all but one of its components smaller than the corresponding components of another probability vector. This theoretical guarantee ensures reduced computational complexity in all real-world tasks considered in this paper. On the other hand, the calculation of the proposed loss only involves a single line code (see Appendix C), making it easy to implement at minimal cost. Moreover, additional experiments highlight the small running time of our method (see above A2). Therefore, the proposed loss is not only easy to implement but also theoretically efficient.

---

> ### Author Response · Authors · 2023-11-18
> **Response 2 to reviewer 29Ss**
>
> >Q4: When there are many classes (such as ImageNet or CIFAR-100), the performance improvement is limited.
>
> A4: Thank you for your comments. First of all, our method achieves SOTA performance across all benchmarks, ranging from small to large datasets. The consistent improvements observed on all datasets strongly emphasize the superiority of our method over previous works.
> Secondly, in terms of ImageNet top-1 accuracy, our method exhibits a 1.28% improvement over the current SOTA method FreeMatch -- equivalent to FreeMatch's improvement over the previous SOTA method FlexMatch. **Notably, such gains on ImageNet are significant, not only due to the dataset's scale but also because our method is both fast and easy to implement.**
> Thirdly,  our loss could be seamlessly inserted into pseudo-label based algorithms as a universal add-on to improve performance (for example FlexMatch, see Appendix D.1). This indicates that our method is ready for integrating with future SOTA methods to achieve additional improvement.
> In summary, while the performance improvement on datasets with many classes may appear smaller than those with a smaller number of classes, our method's performance improvement on datasets with many classes is not limited compared to other strong baselines.

---

> ### Author Response · Authors · 2023-11-20
> **We would be grateful if you could take a look at the response**
>
> Dear Reviewer 29Ss:
>
> We sincerely appreciate your valuable time devoted to reviewing our manuscript. We would like to gently remind you of the approaching deadline for the discussion phase. We have diligently addressed the issues you raised in your feedback, providing detailed explanations. For instance, we explain why our OT loss can be calculated in O(K) in the pesudo-label based semi-supervised learning (which is the scenario we considered). We also a new algorithm block in Appendix C of the revised manuscript to better explain the simplicity of our new OT loss. Would you kindly take a moment to look at it?
>
> We are very enthusiastic about engaging in more in-depth discussions with you.

---

> ### Author Response · Authors · 2023-11-22
> **Seeking Your Input on Revised Paper's Alignment with ICLR Standards**
>
> Dear Reviewer 29Ss,
>
> As the discussion period approaches its conclusion, **we want to ensure that we have thoroughly addressed all your concerns and that our revised paper fully meets the standards of ICLR**. We would highly value any additional feedback you may provide.
>
> Thank you sincerely for your time and consideration.
>
> Best regards,
>
> The Authors

---

### Official Review · Reviewer_BnJk · 2023-10-31

**Soundness:** 2 fair
**Presentation:** 2 fair
**Contribution:** 2 fair
**Rating:** 5
**Confidence:** 5

**Summary:**

This paper proposes a semi-supervised learning method based on optimal transport, exploiting the relation of different classes. Evaluated on the classic benchmark, the proposed method outperforms the previous SOTA methods.

**Strengths:**

1. The proposed method provides state-of-the-art performance on classic benchmarks.
2. Provides a connection between optimal transport and thresholding-based method.

**Weaknesses:**

1. More evaluation results are expected to show on datasets of other modalities [1].
2. As the authors stated, the O(K) complexity of the proposed method comes from the mild assumptions. Providing actual runtime would be helpful to justify this statement further.
3. What's the loss weight of the proposed loss term, How is it affecting training? Ablation study of it is missing.


[1] USB: A Unified Semi-supervised Learning Benchmark for Classification.

**Questions:**

1. The contrastive learning-based methods such as SimMatch and CoMatch also consider the relation between classes. What's difference of optimal transport to this?

---

> ### Author Response · Authors · 2023-11-18
> **Response 1 to reviewer BnJk**
>
> >Q1: More evaluation results are expected to show on datasets of other modalities [1].
>
> A1: Thank you for your suggestion. **In our initial submission, we aligned our experiment settings closely with our main baseline FreeMatch, focusing primarily on image classification for a fair comparative analysis. However, in response to your suggestion, we extend our evaluations to encompass USB [1] of language modality.** These additional experiments validate our previous findings and further affirm our conclusions.
> Specifically, the results below (error rate) demonstrate that on both Amazon Review and Yelp Review, when our approach is integrated with Flex-Match (current state-of-the-art in USB), we have achieved a significant improvement, reaching a new state-of-the-art. This also validates the fact that, beyond the compatibility with FreeMatch, our approach actually can be effortlessly integrated with various existing methods. These supplementary outcomes have been included in the revised version of our papers Appendix D.1.
> |  | amazon_review_250 | amazon_review_1000 | yelp_review_250 | yelp_review_1000 |
> | --- | --- | --- | --- | --- |
> | FlexMatch | 45.75 | 43.14 | 46.37 | 40.86 |
> | FlexMatch + OT | 43.81 | 42.35 | 43.61 | 39.76 |
>
>
> >Q2: As the authors stated, the O(K) complexity of the proposed method comes from the mild assumptions. Providing actual runtime would be helpful to justify this statement further.
>
> A2: Thank you for your valuable feedback. **It's crucial to highlight that, in all our experiments where pseudo-labels are one-hot vectors, the O(K) time complexity of our proposed method strictly holds without any assumptions.** This theoretical guarantee of time complexity distinguishes our approach from others, such as FreeMatch, which lack such assurances. Additionally, in response to your query, we also conduct experiments to measure the actual runtime of our method and results are in Appendix D.3. **We observe that the running time increases by 0.02s compared to FreeMatch during each iteration.** Our findings indicate that the method's runtime closely aligns with that of Freematch, providing practical evidence supporting the efficiency of our method.
>
> >Q3: What's the loss weight of the proposed loss term? How is it affecting training? Ablation study of it is missing.
>
> A3: Thank you for your detailed questions. In our experiments, we set the loss weight as 0.5. **To address your inquiry, we conducted additional experiments specifically to assess how it impacts the model's performance.** The results are presented in the table below, where the loss weight is denoted by $\lambda$.
> | $\lambda$ | 0.3 | 0.4 | 0.5 | 0.6 | 0.7 |
> | --- | --- | --- | --- | --- | --- |
> | Error rate | 4.91 | 4.95 | 4.72 | 4.97 | 4.98 |
>
> From the table, we observe that when lambda is set to 0.5, our methods achieves the best performance. As a reference, other related works such as FreeMatch, the value of the loss weight is much more sensitive to the final performance. We have incorporated this ablation study into the revised version of our paper's Appendix D.2.
>
>
> Further response:
>
> We acknowledge your concerns regarding our paper (i.e., Q1, Q2, and Q3), which primarily focus on the experiments and method performance. **While we have made concerted efforts to address these points, we would like to refocus your attention on the primary contribution of our paper: Our work introduces an optimal transport-based framework that understands many methods, positioning our OTMatch method as a natural progression within this framework.** This approach enables a comprehensive understanding of existing methodologies from an optimal transport-based perspective, constituting the core contribution of our work. We hope this can address your concerns, and provide a clearer demonstration of the value of our work.

---

> ### Author Response · Authors · 2023-11-18
> **Response 2 to reviewer BnJk**
>
> >Q4: The contrastive learning-based methods such as SimMatch and CoMatch also consider the relation between classes. What's difference of optimal transport to this?
>
> A4: Thank you for your question. While both our OTMatch and contrastive learning-based methods consider the relationship between classes, there are some crucial distinctions. **Our OTMatch focuses on aligning the class classification probabilities of two augmented views, following the line of work such as FixMatch, FlexMatch, and FreeMatch.** In contrast, contrastive learning-based methods emphasize the consistency between two batches of augmented views. To be more specific, our OTMatch calculates each optimal transport loss exclusively involving the two augmented views. In contrast, contrastive learning-based methods such as SimMatch and CoMatch align the two batches by utilizing the representation similarity between samples in the batch. As a result, contrastive learning-based methods necessitate an additional branch, apart from the one calculating sample-wise consistency. Therefore, our OTMatch is orthogonal to contrastive learning-based methods and can be combined with them. We have included citations for SimMatch and CoMatch in the revised version along with the above detailed discussion in Related work and Appendix B.1.

---

> ### Author Response · Authors · 2023-11-20
> **We would be grateful if you could take a look at the response**
>
> Dear Reviewer BnJk:
>
> We sincerely appreciate your valuable time devoted to reviewing our manuscript. We would like to gently remind you of the approaching deadline for the discussion phase. We have diligently addressed the issues you raised in your feedback, providing detailed explanations. For instance, we have added experiments on language modality based on USB as you required. The inclusion of these supplementary experiments can further validate our earlier discoveries and reinforce our conclusions. Notably, the presented results (in terms of error rate) provide evidence that our approach, when combined with FlexMatch (the current leading method in USB), exhibits a substantial improvement. This achievement establishes a new state-of-the-art performance level for both the Amazon Review and Yelp Review datasets. Would you kindly take a moment to look at it?
>
> We are very enthusiastic about engaging in more in-depth discussions with you.

---

> > ### Comment · Reviewer_BnJk · 2023-11-20
> >
> > Thanks for the responses from the authors. My concerns have been largely resolved, and I increased my rating correspondingly.

---

> ### Author Response · Authors · 2023-11-21
> **Thank you for increasing the score**
>
> Dear Reviewer BnJk,
>
> Thank you very much for taking the time to re-evaluate our paper after considering our rebuttal. We greatly appreciate your positive remarks regarding the SOTA method we introduced and our contribution to the connection between optimal transport and thresholding-based methods.
> We are also grateful for the improved score you've given our work. Your constructive feedback and acknowledgment of our efforts is truly encouraging. We assure you that the valuable suggestions and insights from you and other reviewers will certainly be integrated into our revised version.

---

### Author Response · Authors · 2023-11-18
**General response**

We would like to thank the reviewers and area chairs for taking their time to review our paper. Our general response can be summarized as follows:

1. We would like to emphasize that improving semi-supervised learning with a new optimal transport-based loss is only one of our contributions. We have also provided an optimal transport based framework to understand the existing method FreeMatch, making our OTMatch method a natural method from a unified optimal transport-based view. But unfortunately, none of the reviewers have summarized the latter contribution in their summary of our work.

2. We have added experiments including ablation studies on hyperparameter/running time and additional empirical results on more benchmarks in Appendix D.

3. We have summarized our algorithm in Appendix C.

---

### Meta-Review · Area_Chair_kZUt · 2023-12-11

**Metareview:**

This paper considers a semi-supervised learning framework that proposes a new algorithm called OTMatch.
Their goal is to capture the relationships between classes by using optimal transport distance.

While the rebuttal have addressed some of the concerns of the reviewers, most of them think that the paper needs
a major revision/clarification before acceptance.

**Justification For Why Not Higher Score:**

There exists several points that need to be addressed (clarity, positioning wrt to recent state of the art, ablation, ... )

**Justification For Why Not Lower Score:**

n/a

---

### Decision · Program_Chairs · 2024-01-16

Reject